# IQOS Use and Interest by Sociodemographic and Tobacco Behavior Characteristics among Adults in the US and Israel

**DOI:** 10.3390/ijerph20043141

**Published:** 2023-02-10

**Authors:** Hagai Levine, Zongshuan Duan, Yael Bar-Zeev, Lorien C. Abroms, Amal Khayat, Sararat Tosakoon, Katelyn F. Romm, Yan Wang, Carla J. Berg

**Affiliations:** 1Braun School of Public Health and Community Medicine, Faculty of Medicine, The Hebrew University of Jerusalem and Hadassah, Jerusalem 9112002, Israel; 2Milken Institute School of Public Health, George Washington University, Washington, DC 20052, USA; 3TSET Health Promotion Research Center, Department of Pediatrics, University of Oklahoma Health Sciences Center, Oklahoma City, OK 73104, USA

**Keywords:** tobacco use, tobacco control, heated tobacco products, global health, alternative tobacco products, non-cigarette tobacco products

## Abstract

Heated tobacco products (HTPs) have expanded globally. IQOS, a global HTP leader, was launched in Israel in 2016 and the US in 2019. To inform tobacco control efforts, it is critical to understand who is likely to use HTPs in different countries with distinct regulatory and marketing contexts. Thus, we conducted a cross-sectional survey among adult (ages 18–45) online panelists in the US (*n* = 1128) and Israel (*n* = 1094), oversampling tobacco users, in the fall of 2021, and used multivariable regression to identify correlates of (1) ever using IQOS; (2) past-month vs. former among ever users; and (3) interest in trying IQOS among never users. Among US adults, correlates of ever use included being Asian (aOR = 3.30) or Hispanic (aOR = 2.83) vs. White, and past-month use of cigarettes (aOR = 3.32), e-cigarettes (aOR = 2.67), and other tobacco (aOR = 3.34); in Israel, correlates included being younger (aOR = 0.97), male (aOR = 1.64), and cigarette (aOR = 4.01), e-cigarette (aOR = 1.92) and other tobacco use (aOR = 1.63). Among never users, correlates of greater interest included cigarette and e-cigarette use in the US (β = 0.57, β = 0.90) and Israel (β = 0.88, β = 0.92). IQOS use prevalence was low (US: 3.0%; Israel: 16.2%) but represented in vulnerable subpopulations (younger adults, racial/ethnic minorities).

## 1. Introduction

Globally, the diversity of tobacco products available and tobacco industry marketing strategies has increased. To inform tobacco control efforts, it is critical to understand who is likely to use different tobacco products in different countries because how various products are regulated and how the industry markets them likely differs across countries [1,2]. Better understanding these differences has been called out as a key priority to advance tobacco control efforts [1,2].

One important area of such investigation is heated tobacco product (HTP) marketing, regulation, and population impact. Interest in HTPs (electronic tobacco products that heat tobacco) has expanded [3,4]. HTPs are available in >60 countries, with considerable uptake in countries such as Japan and Italy, where HTPs have been available since 2014 [3]. Some HTP brands from big tobacco companies include “Mok” (China Tobacco), “iFuse” and “glo” (British American Tobacco), “Ploom TECH” (Japan Tobacco), and “lil” (Korea Tobacco) [3].

The global HTP leader is IQOS from Philip Morris (PM). IQOS is a highly relevant and timely tobacco product for 2 countries—Israel and the US—where IQOS has faced particularly interesting histories of tobacco regulation. Israel represents one of the first countries in which IQOS emerged (December 2016). Initially, IQOS was not regulated as a tobacco product, but since April 2017, it has been—first facing relatively weak tobacco legislation and then more progressive legislation, including an advertisement ban on all tobacco and nicotine products (effective March 2019), a requirement for plain packaging, and prohibited point-of-sale displays (effective January 2020) [5]. IQOS is currently (as at 2022) the only HTP available in Israel [5].

In the US, IQOS emerged in October 2019. In July 2020, the Food and Drug Administration (FDA) responded to IQOS’ Modified Risk Tobacco Products (MRTP) application, authorizing the use of “reduced exposure” (but denying “reduced risk”) claims in its marketing, which included such language as “switching completely” to suggest that changing from cigarettes to IQOS reduces exposure to harmful or potentially harmful chemicals [6]. By May 2021, IQOS was sold in four US states (i.e., Georgia, North Carolina, South Carolina, Virginia) [7]. However, British American Tobacco then pursued a patent-infringement lawsuit against PM that ultimately required PM to discontinue IQOS sales in the US in November 2021 [8]. Nevertheless, PM will likely pursue routes to resume IQOS sales in the US, where the HTP market has potential, [9].

Several aspects of IQOS marketing are noteworthy. First, PM uses innovative methods to promote IQOS, including via novel settings [10,11,12], large point-of-sale displays [12,13], and bold promotional activities [10,11]. Second, IQOS ads promote claims that IQOS (vs. cigarettes) is a “cleaner,” “reduced-risk” product [14,15,16], more acceptable to non-smokers [10,16], and a satisfactory alternative to cigarettes (despite mixed findings [16,17]). In the US, the “reduced exposure” marketing claims authorized by the FDA [18,19] may be misinterpreted as reduced risk [20,21,22]. Moreover, PM has exploited FDA’s authorization to promote IQOS globally, and to influence government regulation of IQOS [15,23]. Leveraging the WHO’s Framework Convention on Tobacco Control as a basis, several countries have imposed strict regulations on IQOS marketing and sales or banned IQOS [23].

A critical component of marketing is identifying target markets. PM asserts that its targeted market for IQOS is current cigarette smokers [18,24,25]. Indeed, in the US, current smokers have shown greater HTP use [26,27,28]. Research in other countries (e.g., Italy [29] and Korea [30]) indicates that IQOS users are more likely to smoke conventional cigarettes and/or electronic cigarettes (e-cigarettes) [30]; however, this work also found that never smokers are equally or more likely than current smokers to try, or intend to try, IQOS [29].

Regarding other consumer characteristics, PM asserts that young people are not target consumers [18,24,25]. However, a 2021 Israeli survey of adolescents aged 15–17 years indicated that 1.8% reported that their first tobacco/nicotine experimentation was with IQOS [31]. Furthermore, a sizeable proportion of US youth and young adults (38.6%) [28] are interested in IQOS; young adults are more likely to use HTPs [26], and youth and young adults are targeted by IQOS marketing [32,33]. In addition, men, racial/ethnic minorities, and lower-income groups may be more likely to use HTPs [9,27,34]. These findings underscore the need to assess subgroups disproportionately impacted by tobacco and historically targeted by the industry [35].

Given the prominence of IQOS as the HTP leader globally, its unique histories in the US and Israel, and the need for cross-country research regarding the evolving tobacco market to advance tobacco control efforts, especially related to tobacco-related disparities [36], the current study examined population subgroups that may be disproportionately impacted by IQOS/HTP in the US and Israel. Specifically, we used cross-sectional data among US and Israel adults (aged 18–45) to examine sociodemographic and other tobacco/nicotine use characteristics in relation to: (1) ever vs. never use; (2) current (past 30-day) vs. former use among ever users; and (3) interest in trying IQOS among never users among US and Israeli adults.

## 2. Materials and Methods

### 2.1. Data Sources

In October-December 2021, a cross-sectional online survey was conducted in the US and Israel, fielded by Ipsos. Eligibility criteria included: 1) ages 18–45 years (because use of alternative tobacco products is most prevalent among those ≤45 [4,37,38]); and 2) able to speak English (US), or Hebrew or Arabic (Israel); in Israel, an additional criterion was having an Israeli ID. Our target sample size was 2000 total participants (1000/country). We aimed to recruit approximately equal sample sizes of males and females in each country. Given the literature documenting racial/ethnic differences in tobacco use in each country [39,40], the samples were constructed to allow for subgroup analyses. Specifically, racial/ethnic group targets in the US were 45% White, 25% Black, 15% Asian, and 15% Hispanic; in Israel, they were 80% Jewish and 20% Arab. We aimed for 40% tobacco users to ensure a sufficient sample size to examine differences among key demographic groups. These sample composition parameters were intended to achieve ≥80% power (at α = 0.05) to detect small to medium effects in relation to the primary outcomes of interest (IQOS use, interest in using IQOS). The final sample included 2222 participants (US, *n* = 1128; Israel, *n* = 1094). The study received ethical approvals from George Washington University (NCR213416) and Hebrew University (27062021) and adhered to STROBE guidelines.

The survey samples were constructed using somewhat different approaches in the 2 countries due to differences in the availability and nature of survey panels (a common limitation in international research [41,42]). Data from samples recruited via different approaches nonetheless allow for analyses examining sociodemographic and tobacco use correlates of IQOS use and interest in trying IQOS.

*US-based sample.* The US survey was conducted primarily using KnowledgePanel^®^, a probability-based web panel designed to be representative, and where members are recruited via a combination of random digit dialing and address-based sampling. KP members are incentivized by KnowledgePanel^®^ points redeemable for cash (typically ~5000 points—equivalent to USD 5—for completing a 25 min survey). As is standard with KnowledgePanel^®^ surveys, multiple prompts (i.e., on days 3, 6, 14, 21, 28 and 35) were made to encourage participation. Of 4960 panelists recruited, 2397 (48.3%) completed eligibility screening, and 1095 (45.7%) of those eligible completed the survey.

To meet subgroup recruitment targets, Ipsos also collected an opt-in (i.e., off-panel) convenience sample of Asian tobacco users. Individuals were recruited using banner ads, web pages, and e-mail invitations; those who clicked on online ads completed eligibility screening (i.e., gender, race/ethnicity, tobacco use). Of 353 individuals screened, 33 (9.3%) were eligible and completed the survey.

*Israel-based sample.* The Israeli survey was conducted among an opt-in sample, using the same approach specified above. Of 2970 individuals who completed the eligibility screening and were eligible, 1094 (36.8%) completed the survey.

### 2.2. Measures

The survey focused on tobacco use and related factors, with a particular focus on IQOS. The survey was translated into Hebrew and Arabic by a professional translation company (Academic Language Experts), back-translated into English, and then examined by 2 bilingual reviewers to verify comparability across translations. Survey content was parallel across countries/languages, except for specific sociodemographics (i.e., origin, religiosity). Israel-based participants could choose to take the survey in Hebrew or Arabic. The survey took ~25 min to complete.

At the outset of the survey, participants were provided images of each tobacco product category and were instructed, “This survey asks about various types of products. We want to ensure that 3 product categories are clear: (1) Ordinary cigarettes are filled with tobacco, lit with a match or lighter, and burned to produce ashes. They can be factory-made or roll-your-own. (2) Vaping products (sometimes called e-cigarettes) heat a *liquid only*. Vaping products do *not* contain actual tobacco. The liquid often contains nicotine and is often flavored. (3) HTPs heat *actual tobacco* to create an emission that is inhaled. HTPs *always* contain actual tobacco in the form of sticks or capsules, or loose tobacco. Some HTPs may also have liquid, but also contain actual tobacco.”

*HTP and other tobacco/nicotine product use.* Participants were asked, “In your lifetime, have you ever used: (1) traditional, ordinary cigarettes—including roll-your-own cigarettes/tobacco? (2) e-cigarettes, vaping products or other electronic nicotine delivery devices (excluding IQOS or similar products)? and (3) HTPs, such as IQOS?” (yes vs. no). Among those reporting lifetime use, current use was assessed by asking, “In the past 30 days, how many days have you used [traditional cigarettes, e-cigarettes, HTPs]?” (no = 0; yes ≥ 1). We similarly assessed hookah/waterpipe/nargila, cigar, pipe, and smokeless tobacco use (reported separately and as use of any of the (4)). Variables were created to indicate current (past 30-day) and former (ever but not past 30-day) use of each product to characterize patterns in bivariate analyses; variables indicating current (vs. no current) use of cigarettes, e-cigarettes, and other tobacco products were used for multivariable analyses.

*Interest in trying IQOS.* Participants were asked: “How likely would you be to try IQOS?” (1 = not at all; 2 = a little; 3 = neutral/unsure; 4 = somewhat; 5 = very).

*Sociodemographics.* Sociodemographic factors included: age; gender; sexual orientation (heterosexual, other); race/ethnicity (in the US: White, Black, Asian, Hispanic; in Israel: Jewish, Arab); nativity; educational attainment (<college degree (or other), ≥college degree); household income (US dollars (USD) or New Israeli Shekels (NIS)); employment status (employed, other); relationship status (married/living with partner, other); and children in the home.

### 2.3. Data Analysis

Parallel analyses were conducted for US and Israel. Results from analysis of weighted data are presented, as the intent was to yield results as representative of the US and Israeli adult populations as possible; however, all analyses were also conducted using unweighted data to determine any differences in results. In the US, weighting adjustments were made to compensate for deviations from equal probability sampling and to account for nonresponse, over-sampling of tobacco users and sociodemographic groups, and other sources of non-sampling error. Weights were based on benchmarks from the Current Population Survey (2021 for sociodemographics, 2018–2019 for tobacco use) [39], including gender by age and race/ethnicity (White, Black, Asian, Hispanic); education; household income; census region; and past 30-day tobacco use by gender and race/ethnicity, respectively. In Israel, all eligible individuals (*n* = 1094) were weighted to represent Israeli adults ages 18–45, using benchmarks from Israel’s 2019 Central Bureau of Statistics for sociodemographics [43] (i.e., gender, ethnicity (Jewish, Arab)) and 2020 tobacco use prevalence [40], similar to the approach used for US weights.

First, descriptive and bivariate analyses (Chi-square for categorical variables; *t*-tests or ANOVAs for continuous variables) were conducted to characterize participants in the US and Israel, respectively, based on IQOS use, categorized as: (1) never users; (2) former users (ever users who had not used in the past 30 days); and (3) current (past 30-day) users. In addition, interest in trying IQOS among never users was examined in relationship to participant characteristics using bivariate analyses (*t*-tests or ANOVAs for categorical variables; Pearson correlations for continuous). Second, multivariable binary logistic regression analyses were conducted to examine correlates of ever (former or current) use of IQOS vs. never use, entering sociodemographic variables in block 1 and other tobacco/nicotine product use in block 2. Selection of sociodemographic variables included in models were based on bivariate findings; multicollinearity was also assessed (resulting in the inclusion of income vs. education and race/ethnicity vs. nativity). Tobacco/nicotine product use variables included in the models were current cigarette, e-cigarette, and other tobacco use, for parsimony and due to small cell sizes of IQOS users (particularly in the US). Finally, we conducted subgroup analyses examining: (1) correlates of current vs. former use among ever users only (binary logistic regression); and (2) correlates of interest in trying IQOS among never users only (linear regression). Analyses were conducted using Stata 15.0 (College Station, TX, USA: Stata Corp LLC); significance was set at 0.05.

## 3. Results

Table 1 presents weighted data for US participants (N = 1128). US participants were on average 31.99 years old, 50.2% female, 12.2% sexual minority, 56.6% White, 13.9% Black, 7.2% Asian, and 22.4% Hispanic. Weighted current, former, and never IQOS use prevalence rates among US adults were 1.1%, 1.9%, and 97.0%, respectively. Note that 41 (63.1%) of the 65 ever IQOS users were current cigarette smokers (17 former; 7 never), while 31 (91.2%) of the 34 current IQOS users were current cigarette smokers (0 former; 3 never). Weighted mean interest in trying IQOS among never users was 1.37 (SD = 0.95; scale of 1 = not at all to 5 = very).

In the US, bivariate analyses indicated that IQOS use status was associated with age, race/ethnicity, nativity, income, and other tobacco product use (see Table 1 for directionality of associations). Higher interest in trying IQOS was correlated with Black or Hispanic race/ethnicity, US-born nativity, lower educational attainment, lower income, and other tobacco product use.

Table 2 presents weighted data for Israeli participants (N = 1094). Israeli participants were on average 29.85 years old, 49.8% female, 17.8% sexual minority, 23.3% Arab, and 89.3% born in Israel. Weighted current, former, and never IQOS use prevalence rates among Israeli adults were 8.2%, 8.0%, and 83.8%, respectively. Note that 157 (68.6%) of the 229 ever IQOS users were current cigarette smokers (41 former; 31 never), while 110 (80.9%) of the 136 current IQOS users were current cigarette smokers (8 former; 18 never). Weighted mean interest in trying IQOS among Israeli never IQOS users was 1.64 (SD = 1.26; scale of 1 = not at all to 5 = very).

In Israel, IQOS use status was associated with gender, income, and other tobacco product use (see Table 2 for directionality of associations). Higher interest in trying IQOS was correlated with being male, having low and middle income, and other tobacco product use.

Table 3 presents weighted multivariable regression results among US adults. Being Asian (adjusted Odds Ratio (aOR) = 3.30, 95% Confidence Interval (95%CI) = 1.30, 8.36), Hispanic (aOR = 2.83, 95%CI = 1.10, 7.24), and current cigarette (aOR = 3.32, 95%CI = 1.38, 8.01), e-cigarette (aOR = 2.67, 95%CI = 1.13, 6.34), and other tobacco product use (aOR = 3.34, 95%CI = 1.47, 7.58) was associated with ever vs. never IQOS use. Among ever IQOS users, only current cigarette use (aOR = 11.88, 95%CI = 2.52, 56.08) was correlated with current vs. former IQOS use (not shown in tables). Among never IOQS users, lower income (≤ USD 24,999 (β = 0.35, 95%CI = 0.16, 0.55) and USD 25,000–149,999 (β = 0.17, 95%CI = 0.05, 0.28) vs. ≥ USD 150,000), and current cigarette (β = 0.57, 95%CI = 0.33, 0.80) and e-cigarette use (β = 0.90, 95%CI = 0.57, 1.24) were associated with greater interest in trying IQOS (Table 3).

Table 4 presents weighted multivariable regression results among Israeli adults. Being younger (aOR = 0.97, 95%CI = 0.94, 0.99), male (aOR = 1.64, 95%CI = 1.05, 2.57), and current users of cigarettes (aOR = 4.01, 95%CI = 2.49, 6.44), e-cigarettes (aOR = 1.92, 95%CI = 1.15, 3.21), and other tobacco products (aOR = 1.63, 95%CI = 1.03, 2.58) were associated with ever vs. never IQOS use. Among ever IQOS users, current cigarette (aOR= 2.64, 95%CI = 1.01, 6.89) and other tobacco use (aOR = 7.03, 95%CI = 2.75, 18.00) were significantly associated with a high likelihood of current vs. former IQOS use (not shown in tables). Among never IQOS users, current cigarette (β = 0.88, 95%CI = 0.64, 1.12) and e-cigarette use (β = 0.92, 95%CI = 0.62, 1.22) were associated with greater interest in trying IQOS (Table 4).

## 4. Discussion

Globally, awareness and use of IQOS have been steadily increasing, particularly among young adult smokers, but also among non-smokers [4,26,28,44,45,46,47,48]. Current findings estimate the prevalence of ever using IQOS among adults ages 18–45 in the US and in Israel as 3.0% and 16.2%, respectively, and current use as 1.1% and 8.2%, respectively, with use more likely among cigarette and e-cigarette users and among certain sociodemographic groups in each country.

The US estimates of ever use (3.5%, with 1.1% current use and 1.9% former) were comparable in this sample to those found in other US-based studies. In the US, a national representative study in 2020 (after IQOS’ FDA MRTP authorization) estimated that 18.7% of US adults aged 18–34 were aware of HTPs and 3.2% were ever users [9]. These findings are similar to 2020 results from a non-probability sample of US young adults indicating HTP awareness and ever use of 19.1% and 4.1%, respectively, which are also higher than earlier estimates in similar age groups (e.g., 9.7% awareness, 3.5% ever use in 2019 [27]; 7.6% awareness, 1.6% ever use in 2017 [49]; 13.9% awareness, 3.0% ever use in 2016/2017 [26]).

In a national representative 2019 Israeli survey, 1.8% of adults reported currently using either IQOS or e-cigarettes [40]. In other repeated cross-sectional surveys among adults above 22 years of age, 1.7%, 0.4% and 1.4% reported past-month use of IQOS in 2019, 2020, and 2021, respectively [31]. Youth (aged 15–17 years) IQOS experimentation increased from 1.0% in 2019 to 5.6% in 2021; a more than a 4-fold increase in 3 years [31]. Current findings indicated a prevalence of 8.2% current IQOS use and 8.0% former use.

Regression results for both countries indicated that current cigarette and e-cigarette use was correlated with ever use (and current use among ever users), as well as with interest in trying IQOS, aligning with prior research in the US [9,26,34,49] and elsewhere (e.g., Japan, Korea) [50,51]. Also of particular note is that 96.9% of current IQOS users in the US and 77.2% in Israel were also currently using cigarettes, undermining the notion that cigarette smokers will switch completely to IQOS. These findings add to the growing literature on the dual use of IQOS together with other tobacco products, with potential for adverse health effects [52].

Regarding sociodemographic factors, similar to the US-based literature [9,27,34], lower income correlated with interest in trying IQOS among US participants [9,27,34]; however, other research has documented associations with higher income [47,53] or no associations [4,37]. Differences across racial/ethnic groups were also found; however, the strength of these associations varied, particularly when accounting for other tobacco use and other sociodemographic characteristics (e.g., income). Indeed, the literature regarding IQOS use across racial/ethnic groups is complex, with some research indicating higher interest and use in non-White adults [26,34] or among White individuals [27], and other research finding no differences [9,49]. While analysis of the US sample found that IQOS use was not related to sex or age, analysis of the Israeli sample documented that being male and younger were correlates of ever using IQOS—which aligns with the literature documenting these associations [9,26,27,34,50,51].

### 4.1. Implications for Policy

Assessing the impact of a new tobacco product such as IQOS is a major challenge, both in foresight and hindsight [54]. The FDA’s MRTP authorization included a requirement that PM conduct surveillance “to determine the impact of the order on consumer perception, behavior, and health…” [6]. However, independent research on IQOS use by population groups is needed to provide the level of rigor necessary to protect public health. While current findings may support the industry claim’s that IQOS is intended to appeal to current cigarette users, our results also reflect several concerns. First, non-smokers indicate some level of IQOS use and interest. Second, almost all current IQOS users in the US and over three-fourths in Israel were also currently smoking cigarettes, undermining the potential harm reduction of IQOS and amplifying concerns regarding increasing dual and polytobacco product use. Finally, given the signals across these samples indicating appeal of IQOS among lower-income populations and racial/ethnic minority groups, there is a need to monitor the impact of IQOS marketing on disadvantaged populations and on health equity [55].

### 4.2. Strengths and Limitations

This study assessed IQOS use and interest in two countries with unique regulatory and marketing settings and involved national samples, methods enabling proper weighting and estimation, and parallel data analyses by country to compare findings. However, current findings have limited generalizability. First, participants were recruited via an online panel in the US and via blended online methods in Israel and for subgroups (Asians) in the US; thus, our recruitment approaches differed across countries and may not have yielded representative or comparable samples, which may have implications for the study findings. Second, there may have been differences between those who participated vs. those who chose not to, and there is limited data available to examine differences between these groups. Third, this sample was restricted to those aged 18–45 (given that HTPs and other alternative tobacco products are most prevalent in this group [4,37,38]); however, whether current findings can be generalized to older or younger populations in these countries requires further research. Fourth, despite attempts to align sociodemographic variables, such as household income and education, there are unavoidable differences in definitions in the US vs. Israel. Fifth, data were self-reported, and thus subject to bias reporting, and the cross-sectional design limits our ability to infer causation, temporality, and interest in trying IQOS in predicting use. Lastly, the low IQOS use rates, particularly current use in the US, undermine statistical power for certain analyses (e.g., subgroup analyses).

## 5. Conclusions

Cross-country research is critical to advance tobacco control efforts, especially as new tobacco products evolve and emerge. In this sample of US adults, IQOS use is not negligible, with possibly increased prevalence among specific populations such as Asians and Hispanics and greater interest among lower-income populations. In this Israeli sample, we found common current and ever IQOS use rates were higher among younger adults and men. Equally important, while findings indicated that cigarette users (vs. non-users) were more likely to use IQOS, almost all IQOS users in the US and over three-fourths in Israel were also currently using cigarettes, with these individuals and others also reporting use of e-cigarettes and other tobacco products. These findings undermine the notion that people will “switch completely” from cigarettes to IQOS and underscore concerns about IQOS and other tobacco products contributing to increasing dual and polytobacco use globally. Moreover, some never smokers and even more former smokers reported IQOS use and interest in use, highlighting additional concerns about population impact. Thus, surveillance of tobacco industry marketing strategies and of consumer HTP perceptions and behavior, particularly among population subgroups (i.e., minorities, low-income, younger adults), is critical.

## Figures and Tables

**Table 1 ijerph-20-03141-t001:** Weighted US participant characteristics (N = 1128) and bivariate analyses examining correlates of never, former, and current IQOS use (weighted IQOS use prevalence of 97.0%, 1.9%, and 1.1%, respectively) and interest in using IQOS in never users.

		IQOS Use Status ^				Interest in Trying	
	Total	Never	Former	Current		Among Never Users *	
	N = 1128	N = 1051	N = 31	N = 36		N = 1051	
Variables	N (%) (or M, SE)	N (%) (or M, SE)	N (%) (or M, SE)	N (%) (or M, SE)	*p*	M (SE) (or r)	*p*
** *Sociodemographics* **							
Age, M (SE) and r	31.99 (0.32)	31.94 (0.33)	33.45 (2.15)	31.75 (1.73)	**<0.001**	−0.01	0.859
Gender							
Male	566 (49.8)	528 (50.1)	14 (38.3)	21 (59.6)	0.377	1.40 (0.04)	0.354
Female	562 (50.2)	523 (49.9)	17 (61.7)	15 (40.4)		1.34 (0.04)	
Sexual orientation							
Heterosexual	983 (87.8)	917 (87.9)	27 (81.4)	31 (85.6)	0.711	1.35 (0.03)	0.085
Other	144 (12.2)	133 (12.1)	4 (18.6)	5 (14.4)		1.52 (0.09)	
Race/ethnicity							
White	493 (56.6)	473 (57.6)	11 (31.2)	6 (35.2)	**0.004**	1.36 (0.04)	**<0.001**
Black	284 (13.9)	265 (13.7)	6 (11.1)	10 (23.2)		1.41 (0.06)	
Asian	177 (7.2)	158 (7.2)	5 (9.4)	14 (13.1)		1.27 (0.05)	
Hispanic	174 (22.4)	155 (21.5)	9 (48.4)	6 (28.4)		1.41 (0.07)	
Nativity							
Born in US	1023 (92.5)	954 (92.6)	26 (81.9)	33 (97.4)	**0.028**	1.38 (0.03)	**0.022**
Born outside of US	105 (7.5)	97 (7.4)	5 (18.1)	3 (2.6)		1.22 (0.06)	
Educational attainment							
<College degree	644 (64.5)	602 (64.4)	17 (60.6)	19 (69.3)	0.843	1.43 (0.04)	**0.004**
≥College degree	484 (35.5)	449 (35.6)	14 (39.4)	17 (30.7)		1.27 (0.04)	
Household income							
≤USD 24,999	183 (9.6)	163 (9.2)	8 (16.7)	10 (28.8)	**0.015**	1.65 (0.09)	**<0.001**
USD 25,000 to 149,999	735 (67.2)	687 (67.2)	19 (73.8)	23 (59.8)		1.40 (0.04)	
≥USD150,000	210 (23.2)	201 (23.6)	4 (9.5)	3 (11.3)		1.18 (0.04)	
Employment status							
Employed	825 (71.1)	779 (71.5)	18 (66.2)	22 (49.6)	0.157	1.35 (0.03)	0.309
Other	303 (28.9)	272 (28.5)	13 (33.8)	14 (50.4)		1.42 (0.06)	
Relationship status							
Married/cohabitating	601 (50.9)	558 (50.8)	19 (63.2)	18 (32.5)	0.187	1.35 (0.04)	0.412
Other	527 (49.1)	493 (49.2)	12 (36.8)	18 (67.5)		1.39 (0.04)	
Children							
Yes	529 (45.8)	487 (45.6)	11 (26.1)	23 (62.3)	0.052	1.39 (0.05)	0.538
No	599 (54.2)	564 (54.4)	20 (73.9)	13 (37.7)		1.35 (0.04)	
** *Tobacco use* **							
Cigarettes							
Never	536 (60.5)	523 (61.9)	4 (14.9)	3 (3.1)	**<0.001**	1.28 (0.04)	**<0.001**
Former	312 (30.0)	294 (29.7)	17 (68.6)	0 (0)		1.30 (0.05)	
Current	253 (9.5)	212 (8.4)	10 (16.6)	31 (96.9)		2.20 (0.11)	
E-cigarettes							
Never	731 (75.0)	713 (76.6)	4 (10.0)	11 (34.0)	**<0.001**	1.24 (0.03)	**<0.001**
Former	198 (17.4)	179 (16.5)	19 (73.5)	0 (0)		1.51 (0.08)	
Current	170 (7.7)	138 (6.9)	7 (16.5)	25 (66.0)		2.40 (0.17)	
Hookah							
Never	761 (75.7)	736 (76.7)	11 (35.0)	11 (33.7)	**<0.001**	1.36 (0.03)	**<0.001**
Former	287 (22.4)	268 (21.9)	17 (60.9)	2 (3.5)		1.35 (0.05)	
Current	62 (2.1)	36 (1.4)	3 (4.1)	23 (62.8)		2.32 (0.34)	
Cigars							
Never	666 (68.3)	641 (69.3)	9 (30.2)	12 (29.6)	**<0.001**	1.31 (0.04)	**<0.001**
Former	327 (27.5)	307 (27.1)	18 (62.1)	2 (5.3)		1.42 (0.05)	
Current	113 (4.3)	87 (3.5)	4 (7.7)	22 (65.1)		1.97 (0.18)	
Pipe							
Never	985 (92.2)	946 (93.7)	16 (46.6)	16 (36.7)	**<0.001**	1.35 (0.03)	**<0.001**
Former	101 (6.7)	83 (5.9)	13 (49.7)	5 (9.1)		1.68 (0.14)	
Current	29 (1.1)	12 (0.4)	2 (3.7)	15 (54.2)		1.74 (0.33)	
Smokeless tobacco							
Never	969 (91.2)	928 (92.3)	19 (52.9)	16 (50.0)	**<0.001**	1.34 (0.03)	**<0.001**
Former	103 (7.2)	89 (6.6)	10 (42.3)	4 (6.1)		1.60 (0.13)	
Current	43 (1.6)	26 (1.1)	2 (4.8)	15 (43.9)		2.09 (0.33)	
Other #							
Never	563 (61.4)	548 (62.5)	8 (27.2)	4 (18.5)	**<0.001**	1.30 (0.04)	**<0.001**
Former	363 (32.2)	346 (32.1)	16 (59.8)	1 (0.6)		1.36 (0.05)	
Current	168 (6.4)	130 (5.4)	7 (13.0)	31 (80.9)		2.02 (0.15)	

Bold indicates significance. ^ Unweighted numbers shown (see table title for weighted IQOS use prevalence estimates). Missing response for IQOS use status, *n* = 10. * Participants were asked: “How likely would you be to try IQOS?” (1 = not at all, 2 = a little, 3 = neutral/unsure, 4 = somewhat, 5 = very). # Other tobacco included hookah, cigars, pipe, and smokeless tobacco. Frequencies of missing data: IQOS use (*n* = 10), sexual orientation (*n* = 1), cigarettes (*n* = 27), e-cigarettes (*n* = 29), hookah (*n* = 18), cigars (*n* = 22), pipes (*n* = 13), smokeless (*n* = 13), other tobacco (*n* = 34).

**Table 2 ijerph-20-03141-t002:** Weighted Israeli participant characteristics (N = 1094) and bivariate analyses examining correlates of never, former, and current IQOS use (weighted IQOS use prevalence of 83.8%, 8.0%, and 8.2%, respectively) and interest in using IQOS in never users.

		IQOS Use Status ^				Interest in Trying	
	Total	Never	Former	Current		Among Never Users *	
	N = 1094	N = 863	N = 93	N = 136		N = 863	
Variables	N (%) (or M, SE)	N (%) (or M, SE)	N (%) (or M, SE)	N (%) (or M, SE)	*p*	M (SE) (or r)	*p*
** *Sociodemographics* **							
Age, M (SE) and r	29.85 (0.27)	30.06 (0.30)	28.23 (0.82)	29.22 (0.84)	**<0.001**	0.04	0.191
Gender							
Male	538 (50.2)	393 (47.3)	59 (62.9)	85 (67.4)	**<0.001**	1.78 (0.06)	**0.001**
Female	556 (49.8)	470 (52.7)	34 (37.1)	51 (32.6)		1.51 (0.05)	
Sexual orientation							
Heterosexual	901 (82.2)	710 (82.2)	84 (88.9)	106 (77.0)	0.231	1.63 (0.04)	0.904
Other	193 (17.8)	153 (17.8)	9 (11.1)	30 (23.0)		1.65 (0.11)	
Race/ethnicity							
Jewish	954 (76.7)	754 (77.1)	80 (74.4)	118 (74.2)	0.853	1.60 (0.04)	0.168
Arab	140 (23.3)	109 (22.9)	13 (25.6)	18 (25.8)		1.77 (0.12)	
Nativity							
Born in Israel	965 (89.3)	758 (89.0)	84 (93.0)	121 (89.1)	0.490	1.65 (0.04)	0.471
Born outside of Israel	129 (10.7)	105 (11.0)	9 (7.0)	15 (10.9)		1.56 (0.10)	
Educational attainment							
<College degree	470 (42.2)	378 (41.3)	40 (45.4)	51 (47.4)	0.523	1.68 (0.06)	0.360
≥College degree	624 (57.8)	485 (58.7)	53 (54.6)	85 (52.6)		1.61 (0.05)	
Household income							
≤ NIS 30,000	204 (24.4)	155 (23.8)	18 (29.3)	30 (24.2)	0.527	1.62 (0.10)	**<0.001**
NIS 30,001–192,000	569 (59.4)	439 (59.2)	47 (55.1)	83 (65.8)		1.80 (0.06)	
≥ NIS 192,001	150 (16.2)	121 (17.0)	15 (15.7)	14 (10.0)		1.42 (0.08)	
Employment status							
Employed	764 (66.9)	605 (68.7)	62 (57.7)	97 (59.6)	0.087	1.65 (0.05)	0.743
Other	330 (33.1)	258 (31.3)	31 (42.3)	39 (40.4)		1.62 (0.08)	
Relationship status							
Married/cohabitating	585 (49.3)	454 (49.5)	48 (46.6)	82 (49.8)	0.895	1.70 (0.05)	0.109
Other	509 (50.7)	409 (50.5)	45 (53.4)	54 (50.2)		1.57 (0.06)	
Children							
Yes	596 (53.4)	473 (54.7)	51 (49.8)	72 (45.3)	0.260	1.58 (0.05)	0.116
No	498 (46.6)	390 (45.3)	42 (50.2)	64 (54.7)		1.71 (0.06)	
** *Tobacco use* **							
Cigarettes							
Never	484 (57.2)	452 (65.1)	13 (15.8)	18 (16.7)	**<0.001**	1.38 (0.04)	**<0.001**
Former	180 (19.2)	139 (17.6)	33 (49.5)	8 (6.1)		1.47 (0.07)	
Current	428 (23.6)	271 (17.3)	47 (34.7)	110 (77.2)		2.73 (0.10)	
E-cigarettes							
Never	659 (69.6)	602 (77.7)	17 (21.5)	40 (34.6)	**<0.001**	1.39 (0.04)	**<0.001**
Former	157 (13.2)	99 (9.7)	47 (57.2)	11 (5.7)		1.97 (0.13)	
Current	275 (17.2)	160 (12.5)	29 (21.3)	85 (59.7)		2.91 (0.14)	
Hookah							
Never	560 (57.7)	485 (62.3)	17 (23.1)	58 (46.0)	**<0.001**	1.46 (0.04)	**<0.001**
Former	295 (24.4)	221 (22.2)	57 (62.4)	16 (8.9)		1.69 (0.07)	
Current	238 (17.8)	157 (15.5)	19 (14.6)	62 (45.1)		2.25 (0.14)	
Cigars							
Never	790 (75.1)	692 (82.8)	25 (25.6)	72 (45.8)	**<0.001**	1.52 (0.04)	**<0.001**
Former	187 (16.9)	113 (12.1)	61 (69.4)	12 (14.1)		1.95 (0.14)	
Current	116 (8.0)	57 (5.1)	7 (5.0)	52 (40.1)		2.76 (0.18)	
Pipe							
Never	900 (84.1)	779 (91.4)	44 (45.0)	76 (47.0)	**<0.001**	1.58 (0.04)	**<0.001**
Former	103 (9.4)	46 (5.0)	44 (50.9)	12 (12.9)		1.91 (0.18)	
Current	90 (6.6)	37 (3.5)	5 (4.1)	48 (40.1)		2.83 (0.23)	
Smokeless tobacco							
Never	896 (83.7)	771 (90.7)	46 (43.3)	78 (52.2)	**<0.001**	1.59 (0.04)	**<0.001**
Former	107 (9.8)	55 (5.7)	40 (51.5)	11 (10.2)		1.81 (0.15)	
Current	91 (6.5)	37 (3.6)	7 (5.3)	47 (37.6)		2.49 (0.23)	
Other #							
Never	449 (47.3)	424 (55.2)	5 (4.1)	20 (10.0)	**<0.001**	1.41 (0.04)	**<0.001**
Former	324 (28.7)	245 (25.6)	63 (76.7)	15 (12.8)		1.64 (0.07)	
Current	319 (24.0)	193 (19.3)	25 (19.2)	101 (77.1)		2.29 (0.12)	

Bold indicates significance. ^ Unweighted numbers shown (see table title for weighted IQOS use prevalence estimates). Missing response for IQOS use status, *n* = 2. * Participants were asked: “How likely would you be to try IQOS?” (1 = not at all, 2 = a little, 3 = neutral/unsure, 4 = somewhat, 5 = very). # Other tobacco included hookah, cigars, pipe, and smokeless tobacco. Frequency of missing data: IQOS use (*n* = 2), household income (*n* = 171), cigarettes (*n* = 2), e-cigarettes (*n* = 3), hookah (*n* = 1), cigars (*n* = 1), pipes (*n* = 1), other tobacco (*n* = 2).

**Table 3 ijerph-20-03141-t003:** Weighted multivariable binary logistic regression predicting ever (vs. never) IQOS use among US adults and multivariable linear regression predicting interest in using IQOS among never users.

	Ever Users vs. Non-Users						Interest in Trying Among Never Users					
	Block 1: Sociodemographics			Block 2: Sociodemographics and Tobacco Use			Block 1: Sociodemographics			Block 2: Sociodemographics and Tobacco Use		
Variables	aOR	95% CI	*p*	aOR	95% CI	*p*	β	95% CI	*p*	β	95% CI	*p*
** *Sociodemographics* **												
Age	1.03	0.98–1.08	0.196	1.03	0.97–1.09	0.420	0.00	0.00–0.01	0.309	0.01	0.00–0.01	0.133
Male (ref: female)	0.79	0.42–1.48	0.464	0.67	0.33–1.37	0.274	0.07	−0.04–0.18	0.234	0.04	−0.07–0.16	0.452
Race/Ethnicity (ref: White)												
Black	1.91	0.78–4.70	0.159	1.39	0.55–3.49	0.488 ^	−0.02	−0.17–0.12	0.761	−0.01	−0.13–0.12	0.898
Asian	2.61	1.12–6.07	**0.026**	3.30	1.30–8.36	**0.012 ^**	−0.04	−0.17–0.09	0.583	0.05	−0.10–0.20	0.514
Hispanic	3.49	1.57–7.76	**0.002**	2.83	1.10–7.24	**0.031**	0	−0.16–0.17	0.966	0.05	−0.11–0.20	0.564
Household income (ref: ≥USD 150,000)												
≤USD 24,999	3.10	1.00–9.64	0.051	3.26	0.92–11.51	0.067	0.48	0.28–0.68	**<0.001**	0.35	0.16–0.55	**<0.001**
USD 25,000 to 149,999	1.56	0.66–3.71	0.313	1.69	0.63–4.57	0.299	0.22	0.11–0.34	**<0.001**	0.17	0.05–0.28	**0.004**
** *Current use* **												
Cigarettes (ref: no)				3.32	1.38–8.01	**0.008**				0.57	0.33–0.80	**<0.001**
E-cigarettes (ref: no)				2.67	1.13–6.34	**0.026**				0.90	0.57–1.24	**<0.001**
Other * (ref: no)				3.34	1.47–7.58	**0.004**				0.14	−0.12–0.41	0.283

Bold indicates significance. * Other tobacco included hookah, cigars, pipe, and smokeless tobacco. ^ In unweighted regression analyses, findings were similar, except that being Black correlated with ever using IQOS and being Asian did not. Thus, racial/ethnic differences are interpreted with caution.

**Table 4 ijerph-20-03141-t004:** Weighted multivariable binary logistic regression predicting ever (vs. never) IQOS use among Israeli adults and multivariable linear regression predicting interest in using IQOS among never users.

	Ever Users vs. Non-Users						Interest in Trying Among Never Users					
	Block 1: Sociodemographics			Block 2: Sociodemographics and Other Tobacco Use			Block 1: Sociodemographics			Block 2: Sociodemographics and Other Tobacco Use		
Variables	aOR	95% CI	*p*	aOR	95% CI	*p*	β	95% CI	*p*	β	95% CI	*p*
** *Sociodemographics* **												
Age	0.97	0.94–0.99	**0.010**	0.97	0.94–0.99	**0.018**	0.00	−0.01–0.01	0.757	0.00	0.00–0.01	0.328
Male (ref: female)	2.01	1.35–3.00	**0.001**	1.64	1.05–2.57	**0.029**	0.23	0.05–0.41	**0.014**	0.11	−0.04–0.26	0.162
Jewish (ref: Arab)	1.01	0.58–1.77	0.970	0.96	0.52–1.77	0.893	−0.24	−0.53–0.06	0.114	−0.21	−0.44–0.02	0.070
Household income (ref: ≥ NIS 192,001)												
≤NIS 30,000	1.35	0.72–2.55	0.350	1.13	0.58–2.21	0.726	0.20	−0.06–0.45	0.133	0.10	−0.11–0.32	0.355
NIS 30,001–192,000	1.22	0.71–2.08	0.475	0.92	0.53–1.59	0.750	0.36	0.16–0.56	**<0.001**	0.16	−0.01–0.34	0.062
** *Current use* **												
Cigarettes (ref: no)				4.01	2.49–6.44	**<0.001**				0.88	0.64–1.12	**<0.001**
E-cigarettes (ref: no)				1.92	1.15–3.21	**0.013**				0.92	0.62–1.22	**<0.001**
Other tobacco * (ref: no)				1.63	1.03–2.58	**0.038**				0.16	−0.08–0.41	0.195

Bold indicates significance. * Other tobacco included hookah, cigars, pipe, and smokeless tobacco.

## Data Availability

The data presented in this study are available on request from the corresponding author. The data are not publicly available due to ethical reasons.

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
