# Peer review of "IQOS Use and Interest by Sociodemographic and Tobacco Behavior Characteristics among Adults in the US and Israel"

_ijerph, 2023, doi:10.3390/ijerph20043141_

Round 1
Reviewer 1 Report
This paper mainly describes the author conducted a questionnaire survey on the use of IQOS among more than 2000 people in the United States and Israel. And the results of the investigation are analyzed and reported. The topic of this paper has a certain practical significance, the conclusion of the analysis is logical, it is suggested to modify and publish.
Revise opinions:
1. Why are people over 45 not included in the survey?
2. Part 4.2 of the paper, “Prevalence estimates on current users in the US might have been affected by the IQOS ban on sales implemented in end of November 2021.” This remark is not appropriate.
3. “Equally important, while findings indicated that cigarette users (vs. non-users) were more likely to use IQOS, almost all IQOS users in the US and over three-fourths in Israel were also currently using cigarettes.” It is suggested to analyze the reasons for this phenomenon.
4. It is suggested to add some analysis of the phenomena found through statistical data.
5. Please explain the evaluation criteria for SE and size in line 205.
6. In Section 2.1. The source of the data, why is the percentage of the population in each country divided this way?
7. It is recommended that the results section on table analysis be placed behind the table.
Author Response
Reviewer 1:
This paper mainly describes the author conducted a questionnaire survey on the use of IQOS among more than 2000 people in the United States and Israel. And the results of the investigation are analyzed and reported. The topic of this paper has a certain practical significance, the conclusion of the analysis is logical, it is suggested to modify and publish.
Response: Thank you for your thoughtful suggestions; addressing them has enhanced the quality of the manuscript.
- Why are people over 45 not included in the survey?
Response: We appreciate this comment and considered this in constructing this study. The prevalence of alternative tobacco product use dramatically drops above the age of 45, so given the scope/focus of this study, we chose to limit the upper age range to 45. We have noted our rationale for this in the Methods section and this as a Limitation.
- Page 3: “Eligibility criteria included: 1) ages 18-45 years (because use of alternative tobacco products is most prevalent among those ≤45 [1-3]); and 2) able to speak English (US), or Hebrew or Arabic (Israel); in Israel, an additional criterion was having an Israeli ID.”
- Page 11: “Third, this sample was restricted to those ages 18-45 (given that HTPs and other alternative tobacco products are most prevalent in this group [1-3]); however, whether current findings generalize to older or younger populations in these countries requires further research.”
- Part 4.2 of the paper, “Prevalence estimates on current users in the US might have been affected by the IQOS ban on sales implemented in end of November 2021.” This remark is not appropriate.
Response: We have deleted this comment.
- “Equally important, while findings indicated that cigarette users (vs. non-users) were more likely to use IQOS, almost all IQOS users in the US and over three-fourths in Israel were also currently using cigarettes.” It is suggested to analyze the reasons for this phenomenon. It is suggested to add some analysis of the phenomena found through statistical data.
Response: These data are presented in Table 1, which we have more explicitly pointed out in the revised manuscript:
- Page 5: “Table 1 presents weighted data for US participants (N=1,128). US participants were on average 31.99 years old, 50.2% female, 12.2% sexual minority, 56.6% White, 13.9% Black, 7.2% Asian, and 22.4% Hispanic. Weighted current, former, and never IQOS use prevalence rates among US adults were 1.1%, 1.9%, and 97.0%, respectively. Note that 41 (63.1%) of the 65 ever IQOS users were current cigarette smokers (17 former; 7 never); 31 (91.2%) of the 34 current IQOS users were current cigarette smokers (0 former; 3 never)….”
- Page 7: “Table 2 presents weighted data for Israel participants (N=1,094). Israel participants were on average 29.85 years old, 49.8% female, 17.8% sexual minority, 23.3% Arab, and 89.3% born in Israel. Weighted current, former, and never IQOS use prevalence rates among Israeli adults were 8.2%, 8.0%, and 83.8%, respectively. Note that 157 (68.6%) of the 229 ever IQOS users were current cigarette smokers (41 former; 31 never); 110 (80.9%) of the 136 current IQOS users were current cigarette smokers (8 former; 18 never)….”
Further, we go on to interpret why this is important to note (p. 12): “These findings undermine the notion that people will “switch completely” from cigarettes to IQOS and underscore concerns about IQOS and other tobacco products contributing to increasing dual and polytobacco use globally. Moreover, some never smokers and even more former smokers reported IQOS use and interest in use, highlighting additional concerns about population impact.”
- Please explain the evaluation criteria for SE and size in line 205.
Response: We have clarified the scale used for this variable and have provided SD rather than SE in text of the results:
- Page 5: “Weighted mean interest in trying IQOS among never users was 1.37 (SD=0.95; scale of 1=not at all to 5=very).”
- Page 7: “Weighted mean interest in trying IQOS among Israel never IQOS users was 1.64 (SD=1.26; scale of 1=not at all to 5=very).”
- In Section 2.1. The source of the data, why is the percentage of the population in each country divided this way?
Response: Given the literature documenting racial/ethnic differences in tobacco use in each country, the samples were constructed to allow for subgroup analyses. We have added this in the Methods section (p. 3):
“Our target sample size was 2,000 total participants (1,000/country). We aimed to recruit roughly equal sample sizes of males and females in each country. Given the literature documenting racial/ethnic differences in tobacco use in each country [4, 5], the samples were constructed to allow for subgroup analyses. Specifically, racial/ethnic group targets in the US were: 45% White, 25% Black, 15% Asian, and 15% Hispanic; and in Israel were: 80% Jewish and 20% Arab. We aimed for 40% tobacco users to ensure sufficient sample sizes to examine differences among key demographic groups.”
- It is recommended that the results section on table analysis be placed behind the table.
Response: We have revised the results so the text describing the tables is presented more closely to the respective table.
Reviewer 2 Report
Levine et al. report cross-sectional survey data from online panels in the USA and Israel on IQOS use and interest to do so among adults (18-45 years) in both countries. In general, the manuscript is well-written and I think all surveys generating information on the use of these novel products are important to accumulate information. However, this survey has some major limitations which affects the generalisability of their findings, and although the authors describe the different legislation and tobacco regulation settings in the US and Israel in their Introduction section, the added value and rationale for their choice to compare the two countries in one paper is not entirely clear to me. Particularly, as the sampling methods seem do differ between the countries (Israel: full convenience sample?) so that no statistical comparisons could be made between both countries.
I would like to note that the authors mention most of the main limitations of their study in the limitations section, but I would have appreciated a more extensive discussion of these limitations in the light of their results. I address my minor and major points in my comments to the authors.
Major comments
1. As written above, the specific added value and rationale for the choice to compare the two countries in one paper is not entirely clear to me. This could be pointed out more clearly in the manuscript. Particularly, as the sampling methods seem do differ between the countries (Israel: full convenience sample) so that neither sound/informative nor statistical comparisons could be made between both countries.
2. Generalisability/Representativeness of the samples:
a. It is a major limitation that the survey in Israel was conducted using a convenience sample. The strategy to recruit participants using banner ads, web pages etc. (potentially including product information/pictures of HTPs?) poses a severe risk for selection bias which needs to be in more detail.
b. The same is true for the subsample of Asian tobacco users in the US (convenience sample).
i. Non-response in this convenience sample was ~90%? Did the authors had the chance to conduct any non-response analyses?
ii. I was very surprised by the result that being Asian/Hispanic in the US seem to correlate with ever use of IQOS although controlled for socioeconomic status (income). What can be a convincing rationale behind the result that having Asian/Hispanic roots seem to be associated with ever use of HNB if not the fact that people often live in lower socioeconomic status circumstances?
iii. I can only hypothesise on this but what also surprises me in this regard is the result that income in the US is highly correlated with “Interest in trying IQOS” while it was not found to be associated with “ever use of IQOS”. My hypothesis would be that income and/or education was not measured adequately or the categories included in the regression analysis were not chosen adequately (see also my major comment 3 in this context).
iv. Weighting adjustments were made aiming to compensate for deviations from equal probability sampling but in very small samples or subsamples (Asian subsample n=33) results could also be distorted by applying weighting procedures.
3. Income / education as independent variables: Some information would be helpful whether the used categories for each country represent high/middle/low income. Does a college degree really means the same in both countries? To me, these very important variables seem to be summarised in relatively rough categories. Which might also explain why the author did not find associations with ever IQOS use (univariate analyses) while surveys from other countries usually find strong associations.
4. Response rates in the US were ~45% of eligible persons and in Israel ~37%, which is quite ok for surveys according to my experience. However, if at least some data on person characteristics were available (eligibility criteria?), a non-response analysis should be conducted which would provide important information on the representativeness of the final samples.
Minor comments
1. It is unfortunate that no analysis plan was published prior to the analyses, especially since so many variables were examined.
2. I was wondering that the authors did not report missing data per variable. It looks like all questions were answered by all participants, which is surprising- particularly since questions were asked on sensible topics (e.g., sexual orientation or income). Or were only complete cases used for the analysis? If so, this should be reported as well as the number of missings per variable.
3. Abstract, line 25: aOR of 97.77 must be an error?
4. What was the rationale to use a restricted age range with a maximum of 45 years?
5. Table 3: ORs should be aORs
6. What was the rationale for using weighted data in the regression analyses? There are many methodological discussions (https://www.annualreviews.org/doi/10.1146/annurev-statistics-011516-012958) on whether calculating a regression analysis with weighted or unweighted data (it often depends on the research question). By weighting, one can create a risk of creating spurious associations, since the weights are imposed without regard to the outcome measure. Cases could be "over-represented" when applying the adjustment in a model. Perhaps the authors could at least include a brief paragraph on their rationale for using weighted data.
Author Response
Reviewer 2:
Levine et al. report cross-sectional survey data from online panels in the USA and Israel on IQOS use and interest to do so among adults (18-45 years) in both countries. In general, the manuscript is well-written and I think all surveys generating information on the use of these novel products are important to accumulate information. However, this survey has some major limitations which affects the generalisability of their findings, and although the authors describe the different legislation and tobacco regulation settings in the US and Israel in their Introduction section, the added value and rationale for their choice to compare the two countries in one paper is not entirely clear to me. Particularly, as the sampling methods seem do differ between the countries (Israel: full convenience sample?) so that no statistical comparisons could be made between both countries.
I would like to note that the authors mention most of the main limitations of their study in the limitations section, but I would have appreciated a more extensive discussion of these limitations in the light of their results. I address my minor and major points in my comments to the authors.
Response: We appreciate this comment and carefully considered this in developing the manuscript. As noted by this reviewer, we believe that these initial analyses examining how the 2 samples vary in terms of sociodemographic and other tobacco use correlates of HTP use and intentions to use was critical in understanding country-specific differences in these associations and in assessing how these data could be analyzed together or separately in subsequent analyses. Our rationale for including both sets of analyses in one manuscript is that comparing significant findings among the US sample versus the Israel sample affords us the unique opportunity to look at how these correlates differ in 2 countries surveyed at the same time, using the same measures, and contextualized within their HTP history. Hence, despite acknowledged limitations, the study allows cross-country comparison.
Notably, differences in sampling strategies across countries is quite common – for example, in the International Tobacco Project [6], Global Adult Tobacco Survey [7], and other surveillance studies – simply due to the constraints in availability of data (e.g., nature of census data), similar panels, etc. However, this does not preclude the samples from being analyzed or presented together in the same manuscript and often provides a more meaningful perspective regarding marketing and regulatory context in relation to the findings.
Note that we have revised the manuscript to more clearly articulate: 1) the rationale for our decision to present these data together; and 2) the inherent limitations we chose to accept because of our rationale.
- Page 1: “Globally, the diversity of tobacco products available and tobacco industry marketing strategies has increased. To inform tobacco control efforts, it is critical to understand who is likely to use different tobacco products in different countries because how various products are regulated and how the industry markets them likely differs across countries [8, 9]. Better understanding these differences has been called out as a key priority to advance tobacco control efforts [8, 9]. One important area of such investigation is heated tobacco product (HTP) marketing, regulation, and population impact....”
- Page 3: “The survey samples were constructed using somewhat different approaches in the 2 countries due to differences in the availability and nature of survey panels (a common limitation in international research [6, 7]). Data from samples recruited via different approaches nonetheless allow for analyses examining sociodemographic and tobacco use correlates of IQOS use and interest in trying IQOS.”
Major comments
- As written above, the specific added value and rationale for the choice to compare the two countries in one paper is not entirely clear to me. This could be pointed out more clearly in the manuscript. Particularly, as the sampling methods seem do differ between the countries (Israel: full convenience sample) so that neither sound/informative nor statistical comparisons could be made between both countries.
Response: Thank you for this comment and underscoring our need to clarify our rationale. Please see bullet point above in response to the reviewer’s general feedback.
- Generalisability/Representativeness of the samples:
- It is a major limitation that the survey in Israel was conducted using a convenience sample. The strategy to recruit participants using banner ads, web pages etc. (potentially including product information/pictures of HTPs?) poses a severe risk for selection bias which needs to be in more detail.
- The same is true for the subsample of Asian tobacco users in the US (convenience sample).
- Non-response in this convenience sample was ~90%? Did the authors had the chance to conduct any non-response analyses?
- I was very surprised by the result that being Asian/Hispanic in the US seem to correlate with ever use of IQOS although controlled for socioeconomic status (income). What can be a convincing rationale behind the result that having Asian/Hispanic roots seem to be associated with ever use of HNB if not the fact that people often live in lower socioeconomic status circumstances?
- I can only hypothesise on this but what also surprises me in this regard is the result that income in the US is highly correlated with “Interest in trying IQOS” while it was not found to be associated with “ever use of IQOS”. My hypothesis would be that income and/or education was not measured adequately or the categories included in the regression analysis were not chosen adequately (see also my major comment 3 in this context).
- Weighting adjustments were made aiming to compensate for deviations from equal probability sampling but in very small samples or subsamples (Asian subsample n=33) results could also be distorted by applying weighting procedures.
Response: We appreciate all of these comments and will address them collectively. First, we apologize for the error in reporting and have corrected this on page 3 to say: “Of 353 individuals screened, 33 (9.3%) were eligible and completed the survey” (from what was previously inaccurately written: “Of 353 individuals screened and eligible, 33 (9.3%) completed the survey”). Note that we were not able to assess differences between survey completers and non-completers given data access limitations.
Second, the revised manuscript more clearly states our rationale for using weighted data and that we also ran all analyses with both weighted and unweighted data to determine any differences in results (p. 4):
“Parallel analyses were conducted for US and Israel. Results from analysis of weighted data are presented, as the intent was to yield results as representative of the US and Israeli adult populations as possible; however, all analyses were also conducted using unweighted data to determine any differences in results. In the US, weighting adjustments were made to compensate for deviations from equal probability sampling and to account for nonresponse, over-sampling of tobacco users and sociodemographic groups, and other sources of non-sampling error. Weights were based on benchmarks from the Current Population Survey (2021 for sociodemographics, 2018-2019 for tobacco use) [4] including gender by age and race/ethnicity (White, Black, Asian, Hispanic); education; household income; census region; and past 30-day tobacco use by gender and race/ethnicity, respectively. In Israel, all eligible individuals (n=1,094) were weighted to represent Israeli adults ages 18-45, using benchmarks from Israel’s 2019 Central Bureau of Statistics for sociodemographics [10] (i.e., gender, ethnicity [Jewish, Arab]) and 2020 tobacco use prevalence [5], similar to the approach used for US weights.”
Third, we considered differences in sociodemographic characteristics (e.g., race/ethnicity, income, education) across countries. We agree that findings regarding racial/ethnic differences in IQOS use related outcomes particularly in the US should be interpreted cautiously, based on bivariate findings (Table 1, which shows greater use rates in Black and Asian adults vs. White adults) and the footnote in Table 3, stating that regression results using unweighted data indicated higher ever use among Blacks (vs. Whites) but not Asians, while regression results using weighted data indicated higher ever use among Asians (vs. Whites) but not Blacks. We call out the need for cautious interpretation, and the Discussion section now represents a more cautious interpretation of these findings, as well as those related to income and education (p. 11):
“Regarding sociodemographic factors, similar to the US-based literature [11-13], lower income correlated with interest in trying IQOS among US participants [11-13]; however, other research has documented associations with higher income [14, 15] or no association [1, 3]. Differences across racial/ethnic groups were also found; however, the strength of these associations varied, particularly when accounting for other tobacco use and other sociodemographic characteristics (e.g., income). Indeed, the literature regarding IQOS use across racial/ethnic groups is complex, with some research indicating higher interest and use in non-White adults [11, 16] or among White individuals [13], and other research findings no differences [12, 17]. While analysis of the US sample found that IQOS use was not related sex and age, analysis of the Israeli sample documented that being male and younger were correlates of ever using IQOS – which aligns with the literature documenting these associations [11-13, 16, 18, 19].”
Finally, our Limitations section has been expanded to underscore these issues and others related to generalizability (p. 11-12):
“First, participants were recruited via an online panel in the US and via blended online methods in Israel and for subgroups (Asians) in the US; thus, our recruitment approaches differed across countries and may not have yielded representative or comparable samples, which may have had implications for study findings. Second, there may have been differences between those who participated vs. chose not to, and there is limited data available to examine differences between these groups. Third, this sample was restricted to those ages 18-45 (given that HTPs and other alternative tobacco products are most prevalent in this group [1-3]); however, whether current findings generalize to older or younger populations in these countries requires further research. Fourth, despite attempts to align sociodemographic variables, such as household income and education, there are unavoidable differences in definitions in the US vs. Israel.”
- Income / education as independent variables: Some information would be helpful whether the used categories for each country represent high/middle/low income. Does a college degree really mean the same in both countries? To me, these very important variables seem to be summarised in relatively rough categories. Which might also explain why the author did not find associations with ever IQOS use (univariate analyses) while surveys from other countries usually find strong associations.
Response: Thank you for noting this. First, in constructing our sample and our survey responses for education and income, we consulted census data in both countries, for example, to parallel ranges for low-, medium-, and high-income households, as is commonly done in international surveillance studies (e.g., Global Adult Tobacco Survey [7], International Tobacco Project Survey [6]).
Second, our preliminary analyses examined more granular levels of income and education, and findings using the more granular levels generally paralleled results when the categories were simplified (as presented in the manuscript). Thus, for interpretability, we chose the more simplified presentation. With regard to interpretation, the literature is mixed in terms of the relationship between socioeconomic status (SES) and IQOS use: some studies show greater interest and use is related to higher SES [14, 15], some suggest lower SES [20], and some show no association [1, 3]. Thus, the differences in associations in the US versus Israel add to the mixed literature on this topic. We have written in the Discussion section (p. 11):
“Regarding sociodemographic factors, similar to the US-based literature [11-13], lower income correlated with interest in trying IQOS among US participants [11-13]; however, other research has documented associations with higher income [14, 15] or no association [1, 3]. Differences across racial/ethnic groups were also found; however, the strength of these associations varied, particularly when accounting for other tobacco use and other sociodemographic characteristics (e.g., income). Indeed, the literature regarding IQOS use across racial/ethnic groups is complex, with some research indicating higher interest and use in non-White adults [11, 16] or among White individuals [13], and other research findings no differences [12, 17]. While analysis of the US sample found that IQOS use was not related sex and age, analysis of the Israeli sample documented that being male and younger were correlates of ever using IQOS – which aligns with the literature documenting these associations [11-13, 16, 18, 19].”
Third, we have explicitly noted issues related to population demographic differences in the Limitations section (p.12: “despite attempts to align sociodemographic variables, such as household income and education, there are unavoidable differences in definitions in the US vs. Israel”).
- Response rates in the US were ~45% of eligible persons and in Israel ~37%, which is quite ok for surveys according to my experience. However, if at least some data on person characteristics were available (eligibility criteria?), a non-response analysis should be conducted which would provide important information on the representativeness of the final samples.
Response: We appreciate this comment; however, there is limited data available regarding those eligible who did not fully complete the survey (i.e., only age essentially since the recruitment targets were set by race/ethnicity, sex, and tobacco use status). We have noted this and other issues related to representativeness and generalizability in the Limitations section (pp. 11-12), as noted above.
Minor comments
- It is unfortunate that no analysis plan was published prior to the analyses, especially since so many variables were examined.
Response: We appreciate this comment. Please note that: 1) the analysis plan is directly from our federally-funded grant (US NIH R01CA239178-01A1); and 2) the potential correlates we selected for inclusion in the analyses are supported by the literature regarding correlates of tobacco use in general and – to the extent possible – heated tobacco product use.
- I was wondering that the authors did not report missing data per variable. It looks like all questions were answered by all participants, which is surprising- particularly since questions were asked on sensible topics (e.g., sexual orientation or income). Or were only complete cases used for the analysis? If so, this should be reported as well as the number missing per variable.
Response: The administration of the survey required participants to respond to all questions, questions typically included response options for “other – specify”, as well as “prefer not to answer” for sensitive questions. The vast majority of responses were interpretable within the response categories provided, and as appropriate, “other” responses were recategorized appropriately. We have noted “prefer not to answer” responses and any missing values as footnotes in Tables 1 and 2.
- Abstract, line 25: aOR of 97.77 must be an error?
Response: Thank you for noting this concern. This has been corrected in the revised manuscript. Note that the abstract has been revised to elaborate on our rationale for the cross-country comparison in this study, and thus, this specific finding is not presented in the revised abstract (to provide the necessary space).
- What was the rationale to use a restricted age range with a maximum of 45 years?
Response: We appreciate this comment and considered this in constructing this study. The prevalence of alternative tobacco product use dramatically drops above the age of 45, so given the scope/focus of this study, we chose to limit the upper age range to 45. We have added our rationale in the Methods section and also noted this as a limitation in the Limitations section.
- Page 3: “Eligibility criteria included: 1) ages 18-45 years (because use of alternative tobacco products is most prevalent among those ≤45 [1-3]); and 2) able to speak English (US), or Hebrew or Arabic (Israel); in Israel, an additional criterion was having an Israeli ID.”
- Page 11: “Third, this sample was restricted to those ages 18-45 (given that HTPs and other alternative tobacco products are most prevalent in this group [1-3]); however, whether current findings generalize to older or younger populations in these countries requires further research.”
- Table 3: ORs should be aORs.
Response: Thank you for noting this lack of specificity; we have revised Table 3 to indicate aOR.
- What was the rationale for using weighted data in the regression analyses? There are many methodological discussions (https://www.annualreviews.org/doi/10.1146/annurev-statistics-011516-012958) on whether calculating a regression analysis with weighted or unweighted data (it often depends on the research question). By weighting, one can create a risk of creating spurious associations, since the weights are imposed without regard to the outcome measure. Cases could be "over-represented" when applying the adjustment in a model. Perhaps the authors could at least include a brief paragraph on their rationale for using weighted data.
Response: Thank you for noting this concern. We have included in the Methods section (p. 4):
“Parallel analyses were conducted for US and Israel. Results from analysis of weighted data are presented, as the intent was to yield results as representative of the US and Israeli adult populations as possible; however, all analyses were also conducted using unweighted data to determine any differences in results. In the US, weighting adjustments were made to compensate for deviations from equal probability sampling and to account for nonresponse, over-sampling of tobacco users and sociodemographic groups, and other sources of non-sampling error. Weights were based on benchmarks from the Current Population Survey (2021 for sociodemographics, 2018-2019 for tobacco use) [4] including gender by age and race/ethnicity (White, Black, Asian, Hispanic); education; household income; census region; and past 30-day tobacco use by gender and race/ethnicity, respectively. In Israel, all eligible individuals (n=1,094) were weighted to represent Israeli adults ages 18-45, using benchmarks from Israel’s 2019 Central Bureau of Statistics for sociodemographics [10] (i.e., gender, ethnicity [Jewish, Arab]) and 2020 tobacco use prevalence [5], similar to the approach used for US weights.”
As noted, we conducted all analyses using both weighted and unweighted data, and results were largely similar. As a footnote to Table 3, we noted one difference that this reviewer has noted: “In unweighted regression analyses, findings were similar, except being Black correlated with ever using IQOS and Asian did not. Thus, racial/ethnic differences are interpreted with caution.”